# Functionalized PET Waste Based Low-Cost Adsorbents for Adsorptive Removal of Cu(II) Ions from Aqueous Media

**Oana Ionela Ungureanu [1], Dumitru Bulgariu [2,3], Anca Mihaela Mocanu [4] and Laura Bulgariu [1,***

[1] Department of Environmental Engineering and Management, Cristofor Simionescu Faculty of Chemical Engineering and Environmental Protection, Gheorghe Asachi Technical University of Iasi, 700050 Iasi, Romania; ungureanu.oana15@gmail.com

[2] Department of Geology, Faculty of Geography and Geology, Alexandru Ioan Cuza University of Iaşi, 700506 Iasi, Romania; dbulgariu@yahoo.com

[3] Romanian Academy, Iasi Branch, Geography Group, 700506 Iasi, Romania

[4] Department of Organic Biochemical and Food Engineering, Cristofor Simionescu Faculty of Chemical Engineering and Environmental Protection, Gheorghe Asachi Technical University of Iasi, 700050 Iasi, Romania; ancamocanu@ch.tuiasi.ro

* Correspondence: lbulg@ch.tuiasi.ro

**Abstract:** The widespread use of polyethylene terephthalate (PET) in the packaging industry has led to the discharge of huge amounts of such waste into the environment and is an important source of pollution. Moreover, because the degradation of PET waste requires a very long time (over 180 years), the recycling of this waste is the only solution to reduce environmental pollution in this case. The solution proposed in this study, is the transformation of PET waste into granular adsorbent materials by functionalization with different phenolic compounds (phenol, p-chlor-phenol, and hydroxyquinone), and then their use as adsorbent materials for removing metal ions (ex. Cu(II) ions) from aqueous solutions. The functionalization of PET waste was done with different amounts (2–8 g) of each phenolic compound. The adsorption capacity of obtained materials was tested at different initial Cu(II) ions concentrations, in batch systems, at room temperature (20 ± 1 °C). The experimental results have shown that the adsorbent material obtained by the functionalization of PET waste with 8 g of phenol has the best adsorptive performances (q = 12.80 mg g$^{-1}$) at low initial concentrations of Cu(II) ions, while the adsorbent material obtained by the functionalization of PET waste with 2 g of hydroxyquinone is more efficient in removal of high concentrations of Cu(II) ions (q = 61.73 mg g$^{-1}$). The experimental isotherms were modeled using Langmuir and Freundlich isotherm models, to highlight the adsorptive performances of these new adsorbents and their potential applicability in environmental decontamination processes.

**Keywords:** PET waste; phenolic compounds; adsorbents; Cu(II) ions; aqueous media

---

## 1. Introduction

The removal of metal ions from industrial effluents is a major concern worldwide. Due to their high toxic potential, their tendency to accumulate and their persistence, metal ions are considered persistent pollutants and their concentration in wastewater must be carefully controlled [1,2]. An example are Cu(II) ions, which due to their many industrial uses must be removed from industrial effluents before being discharged into the environment [2]. Among the methods for removing metal ions from aqueous media (such as: chemical precipitation, coagulation, ions exchange, membrane processes, catalytic degradation, etc.) [3,4], adsorption is considered the most viable, due to its efficiency,

simple operation, and low cost. Therefore, finding adsorbent materials that have a low cost of preparation and a high efficiency in retaining metal ions from aqueous media is a current issue for which solutions are still being sought.

Polyethylene terephtalate (PET) is one of the most used synthetic polymers in the actual economy around the world, which is used to produce beverage containers, food packaging, fibers, textile, additives for building materials, etc. [5,6]. These multiple uses of PET are the consequence of its high chemical and thermal stability over a long period of time and low manufacturing costs of this polymer [5,7]. Unfortunately, the use of PET for the manufacture of food packaging and beverage containers has led to the discharge of huge amounts of PET waste into the environment, causing its pollution. This is why in modern society, PET waste is considered one of the biggest pollutants, in terms of environmental protection. In addition, as the degradation of PET waste requires a very long period of time (over 180 years) [8], environmental pollution with this waste can be considered permanent. A last study of the European Commission [9] showed that at the end of 2015, PET waste represents more than 8% by weight and 12% of the volume of solid waste in the world. As the storage or incineration of this non-degradable waste has serious limitations, its recycling is the best possible alternative [10]. Thus, a new industrial activity related to post-consumer PET waste recycling has started to be developed due to the pressure to improve waste management and environmental protection [11,12].

Until now, the most common procedure for recycling PET waste is based on its transformation into fibers or granules, which are then used in various industrial sectors (textiles, construction materials, furniture industry, etc.) [13–16]. Recycling procedures involve only few simple operational steps, such as: washing, grinding, and melting of PET waste, followed by their extrusion. Even if such recycling procedures are very cheap and easy to implement on an industrial scale, they have the disadvantage that they lead to obtaining products (fibers, granules) with low added value. Most of the time, fibers and granules obtained by recycling PET waste are used as raw materials or fillers, and therefore, their selling price is low. Therefore, the recycling industry of PET waste is not always profitable, and this makes the interest of investors in this direction still quite low, and the environmental consequences of PET waste pollution remain serious. In order to avoid all these shortcomings, it is necessary to find other recycling procedures to obtain products with higher added value from PET waste and which are requested on the market. One such trend could be the obtaining adsorbent materials from PET waste, proposed in this study, because it is known that adsorption is the most efficient and versatile technique for removing heavy metal ions, even at very low concentrations [17,18].

Obtaining adsorbent materials from PET waste, which can then be used in treatment of wastewater is based on the following considerations: (i) PET waste is available in large quantities in all regions of the world, (ii) has high chemical, mechanical, and thermal stability for a long time, and is therefore suitable for use on an industrial scale, (iii) allows the valorization of this waste in accordance with the principles of the circular economy [19,20], and (iv) will allow the removal of heavy metals, even at very low concentrations from aqueous media, by a simple and efficient method.

Therefore, the search for new ways of exploiting PET waste, as well as avoiding the production of hazardous materials, has become interesting both theoretically and applicative. For example, Monier and Abdel-Latif (2013) [21] reported the possibility of obtaining chelated fibers prepared with PET fibers waste for the rapid removal of heavy metal ions from water. Under these conditions, the possible use of PET waste as an adsorbent material for the removal of heavy metal ions will allow a new possibility to exploit these wastes and will ensure the very low cost of the adsorbent materials. One problem remains, PET fibers waste has a very low affinity for both heavy metal ions and organic dyes in aqueous solution, as shown in previous studies [22,23], which makes the adsorption processes inefficient. This is a consequence of the fact that PET waste does not have functional groups to which metal ions or organic molecules in the aqueous environment can bind. Therefore, an efficient functionalization of PET waste should involve the dissolution of this material in a suitable solvent, followed by an increase in the number of functional groups.

In this study, three phenolic compounds (phenol, p-chloro-phenol, and hydroxyquinone) were used for the functionalization of PET waste, to obtain new adsorbent materials with applications in environmental remediation. The adsorptive performances of the obtained adsorbent materials was examined as a function of the type and amount of phenolic compound used for functionalization in the removal process of Cu(II) ions from aqueous solution. Different initial Cu(II) ions concentrations were used in all experiments, to highlight the practical applicability of the obtained adsorbent materials. All experimental isotherms were modeled using Langmuir and Freundlich isotherm models, and the isotherm parameters were calculated for each case to highlight the adsorptive performances of these adsorbent materials.

## 2. Materials and Methods

### 2.1. Materials

PET flakes used as raw material (with average area of 0.5 cm$^2$) for obtaining adsorbent materials were purchased from from GreenFiber Company (Iaşi, Romania). Phenol, p-chlor-phenol, and hydroxyquinone (Chemical Company, Iaşi, Romania) were used as received. A stock solution of Cu(II) ions (636 mg Cu(II) L$^{-1}$) was obtained by dissolving copper sulfate in distilled water. All working solutions were obtained by diluting the stock solution. All chemical reagents were of analytical grade and were used without purification.

### 2.2. Adsorbent Materials Preparation and Characterization

The experimental procedure used for the preparation of the adsorbent materials involves the mixing of a constant amount of PET flakes (1 g) with different amounts of phenolic compounds (2–10 g) at different temperatures (150 °C—for phenol and p-chlor-phenol, and 250 °C—for hydroxyquinone) for 2 h. The temperature was selected so as to ensure complete melting of the components in the mixture. After melting, the mixtures were cooled to room temperature, ground to uniform granulation and kept in desiccators at constant humidity. Other details about this experimental procedure and the characterization of obtained materials have been presented in a previous study [24]. The notation of the obtained adsorbent materials was made taking into account all the details of preparation. Thus, for each adsorbent material, was specified both the amount (ex. 2, 3, . . . 8) and the type of phenolic compound (ex. Ph for phenol, Cl-Ph for p-chlor-phenol, or OH-Ph for hydroxyquinone) used for the functionalization of PET waste.

FTIR spectra were recorded for each adsorbent material, using a Bio-Rad FTIR spectrometer (700–3500 cm$^{-1}$ spectral domain, with a resolution of 4 cm$^{-1}$, KBr pellet technique).

### 2.3. Adsorption Experiments

All the adsorption experiments were performed, in batch system, at room temperature (20 ± 1 °C) and at constant contact time (24 h). In all cases, a constant quantity of adsorbent material (0.2 g) was mixed with 25 mL of aqueous solution of Cu(II) ions (initial solution pH = 6.5, initial concentration = 12.7–177.9 mg Cu(II) L$^{-1}$), in 100 mL conical flasks, and intermittent stirred. After filtration, the concentration of Cu(II) ions was analyzed spectrophotometrically (VIS Spectrophotometer YA1407020, rubeanic acid $\lambda$ = 390 nm, 1 cm glass cells, against distilled water).

The performances of obtained adsorbent materials in the removal of Cu(II) ions were examined on the basis of adsorption capacity ($q$, mg g$^{-1}$), calculated from the experimental results, using the equation:

$$q = \frac{(c_0 - c) \cdot (V/1000)}{m}, \tag{1}$$

where $c_0$, c are initial and final concentration of Cu(II) ions in solution (mg L$^{-1}$); V is the volume of solution (mL); $m$ is the mass of adsorbent materials used in experiments (g).

All adsorption experiments were carried out in duplicate and the standard deviation, obtained from ANOVA statistical analysis, were lower than 1.5%.

## 3. Results and Discussion

In evaluating the adsorption performance of PET waste functionalized with phenolic compounds (phenol, p-chlor-phenol, and hydroxyquinone) for the removal of Cu(II) ions from an aqueous solution, three parameters were taken into account: (i) the nature of the phenolic compound, (ii) the amount of phenolic compound used for functionalization of PET waste, and (iii) the initial concentration of Cu(II) ions from the aqueous solution. The detailed analysis of the influence of each of these parameters will provide useful information for establishing the optimal conditions for the functionalization of PET waste, and obtaining efficient adsorbent materials for the removal of Cu(II) ions.

### 3.1. Adsorptive Characteristics of Functionalized PET Waste Materials

Phenolic compounds are among the few chemical compounds that can be used to dissolve PET waste [24]. Therefore, when treating PET waste with such compounds, it is expected that the porosity of the obtained materials to increase, as a result of the dissolution of PET waste followed by their solidification. The selection of these three phenolic compounds, namely: phenol, p-chloro-phenol, and hydroxyquinone (p-hydroxyl-phenol), was done to verify if besides to the physical dissolution of PET waste, the nature of the functional groups of phenolic compounds has a role in defining the adsorptive characteristics of the obtained materials. Therefore, PET waste was mixed with phenolic compounds, in a ratio of 1:2, and the obtained materials were used as adsorbent for the retention of Cu(II) ions from aqueous media. The obtained experimental results are illustrated in Figure 1, compared with un-functionalized PET waste (0-PET).

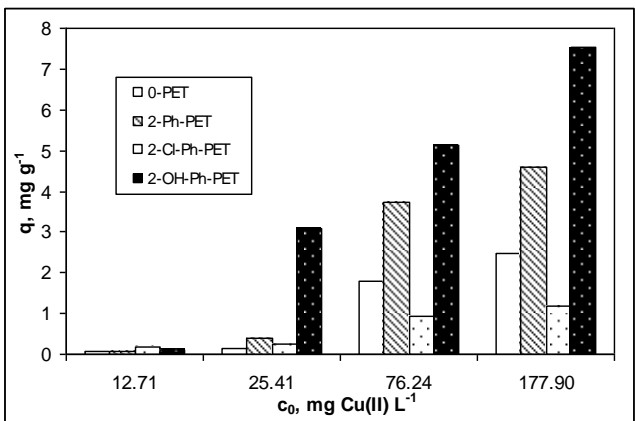

**Figure 1.** Adsorption capacities of functionalized PET waste at different initial Cu(II) ions concentration (pH = 6.5, contact time = 24 h, adsorbent dose = 8 g $L^{-1}$).

It can be observed from Figure 1 that the functionalization of PET waste with phenol and hydroxyquinone leads to obtaining adsorbent materials with higher adsorption capacities than 0-PET, over the entire range of initial Cu(II) ions concentrations. In contrast, in the case of using p-chloro-phenol for the functionalization of PET waste, the obtained material (2-Cl-Ph-PET) has a lower adsorption capacity for Cu(II) ions than unfunctionalized PET waste (0-PET). For the highest initial Cu(II) ions concentration (177.90 mg $L^{-1}$), the adsorption capacity of 2-OH-Ph-PET (7.53 mg $g^{-1}$) is three times higher than the adsorption capacity of 0-PET (2.48 mg $g^{-1}$).

In the case of 2-Ph-PET, the adsorption capacity (4.61 mg $g^{-1}$) is almost two times higher than the adsorption capacity of 0-PET (2.48 mg $g^{-1}$), while in the case of 2-Cl-Ph-PET, the adsorption capacity (1.19 mg $g^{-1}$) is more than two times lower than the adsorption capacity of 0-PET (2.48 mg $g^{-1}$).

Therefore, p-choro-phenol is not suitable for the functionalization of PET waste and has not been used in subsequent experimental studies.

On the other hand, the increase of the adsorption capacity of 2-OH-Ph-PET is notable for much lower initial concentration of Cu(II) ions, than in the case of 2-Ph-PET adsorbent (Figure 1). This suggests that hydroxyquinone is more efficient in the functionalization of PET waste than phenol, and allows to obtain adsorbent materials with better performance for Cu(II) ions from the aqueous solution.

However, before making a decision, the effect of the amount of phenolic compounds (phenol and hydroxyquinone) used for the functionalization of PET waste must also be studied. In this study, the amount of phenolic compounds used for the functionalization of 1 g of PET waste was varied between 2 and 8 g, and the adsorptive performances of the obtained materials for Cu(II) ions from aqueous solution are presented in Figures 2 and 3.

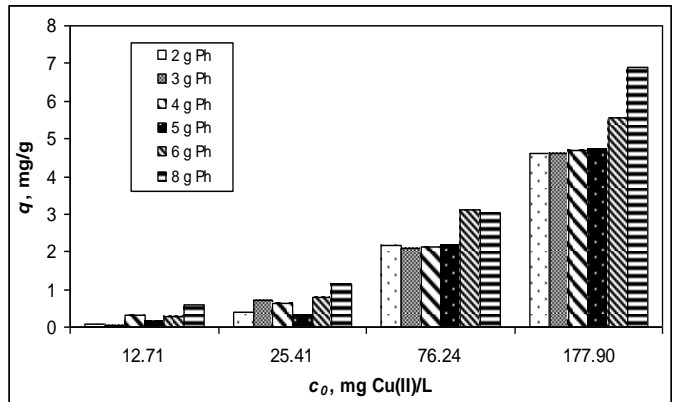

**Figure 2.** Adsorption capacities of functionalized PET waste with different amounts of phenol (Ph), for Cu(II) ions from aqueous media (pH = 6.5, contact time = 24 h, adsorbent dose = 8 g $L^{-1}$).

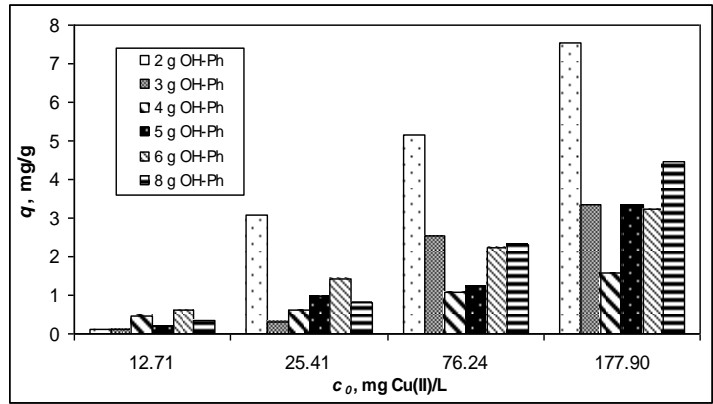

**Figure 3.** Adsorption capacities of functionalized PET waste with different amounts of hydroxyquinone (OH-Ph), for Cu(II) ions from aqueous media (pH = 6.5, contact time = 24 h, adsorbent dose = 8 g $L^{-1}$).

The experimental results from Figure 2 show that in the case of functionalization of PET waste with phenol, the adsorption capacity of the obtained materials increases with the increase of the amount of phenol used for functionalization, and this variation is valid for the entire studied range of initial Cu(II) ion concentration. This increase in adsorption capacities is much more pronounced when the amount of phenol used for functionalization is high, and the concentration of Cu(II) ions in the solution is also high (Figure 2).

For example, at initial Cu(II) ions concentration of 177.90 mg $L^{-1}$, the adsorption capacity of the material obtained by the functionalization of PET waste with 8 g of phenol ($q_{8-Ph-PET}$ = 6.91 mg $g^{-1}$) is with almost 25% higher than the adsorption capacity of material obtained by the functionalization of

PET waste with 6 g of phenol ($q_{6-Ph-PET}$ = 5.53 mg g$^{-1}$), and with more than 46% than the adsorption capacities of the materials obtained by the functionalization of PET waste with 5, 4, 3, and 2 g of phenol.

On the other hand, if the adsorption capacities of 6-Ph-PET and 8-Ph-PET are compared at different initial concentrations of Cu(II) ions, it can be seen that, if at the highest initial Cu(II) ions concentration (177.90 mg L$^{-1}$) the difference between the adsorption capacities is not more than 25%, at the lowest initial Cu(II) ions concentration (12.71 mg L$^{-1}$), the adsorption capacity of 8-Ph-PET is twice the adsorption capacity of 6-Ph-PET (Figure 2).

Therefore, the functionalization of PET waste with a high amount of phenol (8 g) allows to obtain an adsorbent material (8-Ph-PET) which can be effective in removing Cu(II) ions from an aqueous solution, in a wide range of concentrations.

The adsorbent materials obtained by the functionalization of PET waste with hydroxyquinone show a totally different variation of the adsorption capacities (Figure 3). As can be seen from Figure 3, the highest adsorption capacities are obtained in the case of functionalization of PET waste with 2 g of hydroxyquinone (2-OH-Ph-PET), regardless of the initial Cu(II) ions concentration from aqueous solution. In this case, the differences are much more obvious, so that for 2-OH-Ph-PET, the adsorption capacity is over 6% higher for the initial Cu(II) concentration of 12.71 mg L$^{-1}$, and over 68% higher for the initial Cu(II) ions concentration of 177.90 mg L$^{-1}$, compared to other adsorbent materials obtained by the functionalization of PET waste with hydroxyquinone (Figure 3). Therefore, the use of 2 g of hydroxyquinone for the functionalization of PET waste leads to an adsorbent material that can efficiently remove Cu(II) ions from the aqueous solution, especially at high values of their concentration.

All these experimental results indicate that of all the materials obtained by functionalizing PET waste with phenolic compounds, two of them, namely 8-Ph-PET and 2-OH-Ph-PET, have the potential to be used in the processes of removal of metal ions from aqueous solutions. Figure 4 shows a comparison of the adsorbent performances of these two materials for different concentrations of Cu(II) ions, from aqueous solution.

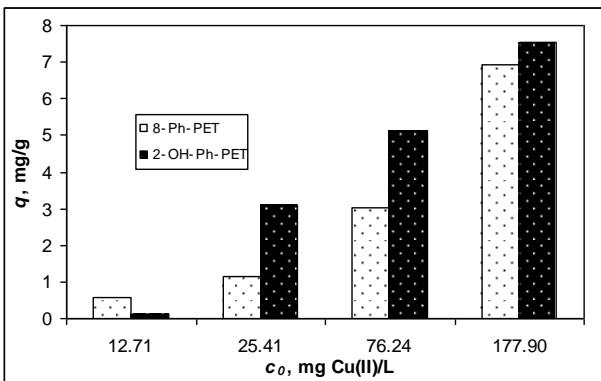

**Figure 4.** Adsorption capacities of 8-Ph-PET and 2-OH-Ph-PET for Cu(II) ions from aqueous media (pH = 6.5, contact time = 24 h, adsorbent dose = 8 g L$^{-1}$).

Although obtaining 8-Ph-PET material requires a large amount of phenol (8 g), which is a very toxic chemical reagent, its adsorptive performances at low initial concentrations of Cu(II) ions (Figure 4) make the modeling of these experimental data necessary, as well.

### 3.2. Modeling of Equilibrium Data

For the quantitative description of the Cu(II) ions, adsorption processes on 8-Ph-PET and 2-OH-Ph-PET, the experimental equilibrium data were modeled using Langmuir and Freundlich isotherm models. The linear equations of these two models [25,26] can be expressed:

$$\text{Langmuir model}: \frac{1}{q} = \frac{1}{q_{max}} + \frac{1}{q_{max} \cdot K_L} \cdot \frac{1}{c}, \qquad (2)$$

$$\text{Freundlich model}: \lg q = \lg K_F + \frac{1}{n} \cdot \lg c, \tag{3}$$

where $q$ is the equilibrium adsorption capacity; $q_{max}$ is maximum adsorption capacity; $K_L$ is Langmuir constant; $KF$ is Freundlich constants; $n$ is heterogenity factor.

The selection of these two models was based on their usefulness in describing the adsorption process. Thus, the Langmuir isotherm model assumes that the adsorption process takes place until a complete monolayer of metal ions is formed on the outer surface of the adsorbent and allows the calculation of the maximum adsorption capacity that corresponds to the surface saturation [27]. The Freundlich model considers that the adsorption process takes place in multiple layers on the heterogeneous surface of the adsorbent material [25], and can be used to estimate the adsorption intensity.

The linear representations of the Langmuir model (1/$q$ vs. 1/c) and Freundlich model (lg $q$ vs. lg $c$) illustrated in Figure 5, have been used for the calculation of isotherm parameters characteristic for the adsorption of Cu(II) ions on 8-Ph-PET and 2-OH-Ph-PET adsorbents. The obtained values of these parameters are summarized in Table 1, together with the regression coefficients ($R^2$), used in the selection of the best model to fit the experimental data.

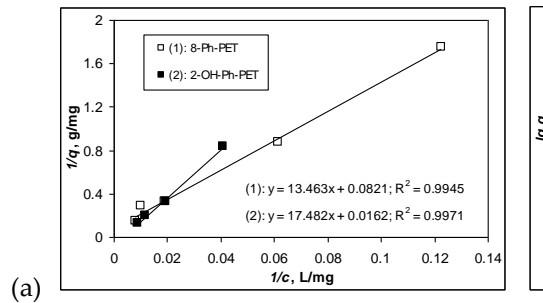 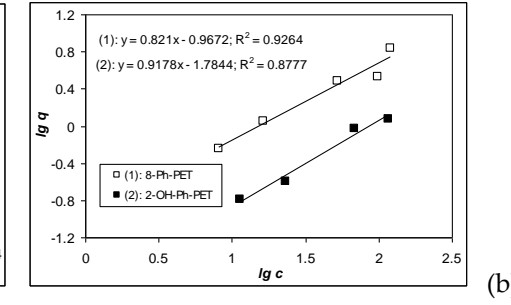

(a) (b)

**Figure 5.** Linear representations of Langmuir (**a**) and Freundlich (**b**) isotherm models for studied adsorption processes.

**Table 1.** Isotherm parameters calculated for the adsorption of Cu(II) ions on 8-Ph-PET and 2-OH-Ph-PET adsorbents.

| Isotherm Model | Parameter | 8-Ph-PET | OH-Ph-PET |
|---|---|---|---|
| Langmuir model | $R^2$ | 0.9945 | 0.9971 |
| | $q_{max}$, mg g$^{-1}$ | 12.80 | 61.73 |
| | $K_L$, L g$^{-1}$ | 0.0061 | 0.0009 |
| Freundlich model | $R^2$ | 0.9264 | 0.8777 |
| | $n$ | 1.22 | 1.09 |
| | $K_F$, L g$^{-1}$ | 0.1078 | 0.0164 |

The correlation coefficients ($R^2$) higher than 0.99 (Figure 5 and Table 1) indicate that the Langmuir model best describes the experimental data obtained on the adsorption of Cu(II) ions on 8-Ph-PET and 2-OH-Ph-PET adsorbents, compared with the Freundlich isotherm model ($R^2 < 0.93$). This means that the adsorption of Cu(II) ions takes place at the surface of the adsorbents and that the maximum adsorption capacity will be determined by the number and availability of surface functional groups to interact with the metal ions in the aqueous solution.

Under these conditions, the maximum adsorption capacity of 2-OH-Ph-PET, which is almost 5 times higher than the maximum adsorption capacity of 8-Ph-PET (Table 1), indicates that the 2-OH-Ph-PET adsorbent has a larger number of functional groups on its surface and is more efficient in adsorption processes. In addition, comparing the values of Langmuir constants (Table 1), it can be said that in the case of 2-OH-Ph-PET, the interaction between Cu(II) ions and functional groups is stronger compared to 8-Ph-PET. Therefore, the functionalization of PET waste with 2 g of hydroxyquinone allows to obtain an adsorbent material much more efficient in the adsorption processes of Cu(II) ions

from aqueous solutions, than the one obtained in the case of functionalization of this waste with 8 g of phenol. These adsorptive performances, together with the non-toxic character of hydroxyquinone and the small amount of reagent required for functionalization, recommend the use of 2-OH-Ph-PET in the removal processes of metal ions from aqueous media.

In order to highlight the potential applicability of these adsorbent materials in the removal processes of Cu(II) ions from aqueous media, the obtained maximum adsorption capacities were compared with the values obtained for other adsorbents reported in literature (Table 2). It should be mentioned that the values presented in Table 2 are the maximum adsorption capacities calculated from the Langmuir isotherm model, under similar experimental conditions.

**Table 2.** Maximum adsorption capacity obtained for the removal of Cu(II) ions from aqueous media using these two materials (8-Ph-PET and 2-OH-Ph-PET) and other adsorbents.

| Adsorbent | pH | Adsorbent Dosage, g L$^{-1}$ | Cu(II) | Reference |
|---|---|---|---|---|
| Montmorillonite clay | 6.0 | 5.0 | 32.26 | [28] |
| Bentonite clay | 6.0 | 5.0 | 10.16 | [29] |
| Cauliflower leaves biochar | 6.0 | 5.0 | 53.96 | [30] |
| Commercial activated carbon | 6.5 | 6.0 | 86.32 | [31] |
| Grape seed activated carbon impregnated with ZnCl$_2$ | 5.0 | 25.0 | 48.78 | [32] |
| Activated carbon of rubber wood sawdust | 6.0 | 5.0 | 6.50 | [33] |
| *Cladyspoium cladosporioides* modified with formaldehyde | 6.0 | 2.0 | 7.74 | [34] |
| *Saccharomyces cerevisiae* modified with methanol | 5.0 | 1.0 | 4.00 | [35] |
| Algal waste | 5.3 | 1.0 | 15.89 | [36] |
| *Ecklonia maxima* modified with CaCl$_2$ | 6.0 | - | 85.00 | [37] |
| Soya been hulls activated with NaOH | 6.0 | 10.0 | 91.51 | [38] |
| 8-Ph-PET | 6.5 | 8.0 | 12.80 | This study |
| 2-OH-Ph-PET | 6.5 | 8.0 | 61.73 | This study |

It can be observed that the maximum adsorption capacity of 2-OH-Ph-PET adsorbent is higher than the values obtained for different clay adsorbents or biochar, and it is comparable with those obtained for commercial activated carbon (Table 2). These encouraging results show that adsorbent materials obtained by functionalization of PET waste with hydroxyquinone have the potential to be used to remove Cu(II) ions from aqueous effluents.

Although the Freundlich model does not describe so well the experimental data obtained at the adsorption of Cu(II) ions on the two adsorbents (Table 1), the detailed analysis of the obtained parameters highlights two very important aspects, namely: (i) the very different values of the Freundlich constant clearly show that the adsorption process of Cu(II) ions takes place with a much higher intensity in the case of 2-OH-Ph-PET than in the case of 8-Ph-PET. This is in agreement with the results obtained in the case of the Langmuir model, and it is an indication that at the functionalization of PET waste with hydroxyquinone, the number of functional groups increases. (ii) The values of the n constant are close for the two adsorbents (Table 1), which shows that the spontaneity of the adsorption process of Cu(II) ions does not change significantly, in function on the nature of the phenolic compound used to functionalize the PET waste. This suggests that, in the structure of the adsorbent materials, the availability of functional groups to interact with Cu(II) ions from aqueous media does not change significantly, in function on the nature of the phenolic compound used to functionalize the PET waste.

In order to clarify these aspects, the FTIR spectra were recorded for each component (PET, 8-Ph-PET and 2-OH-Ph-PET), and their detailed analysis is presented in the next section.

*3.3. FTIR Spectra Analysis*

FTIR spectra have been recorded both for PET waste and for obtained adsorbent materials (8-Ph-PET and 2-OH-Ph-PET), to highlight the structural changes that occur in the PET structure after functionalization. The obtained FTIR spectra are illustrated in Figure 6.

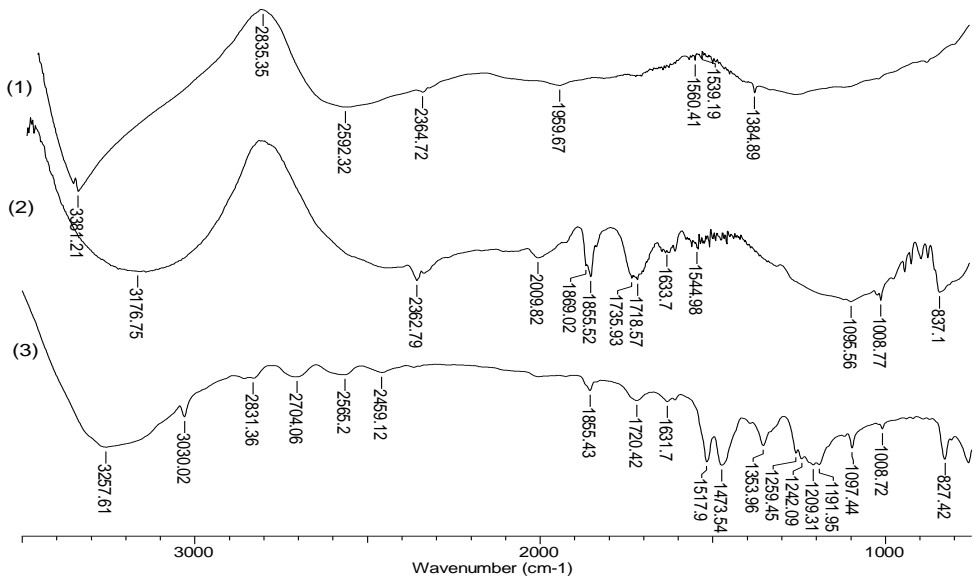

**Figure 6.** FTIR spectra of PET waste (1), 8-Ph-PET (2), and 2-OH-Ph-PET (3).

It can be observed from Figure 6 that in the spectrum of PET waste, (spectrum 1) has a low number of absorption bands (due to its compact structure) and the most intense of these are at 3381, 1560, and 1384 cm$^{-1}$, which can be assigned to the stretching vibrations of terminal OH groups, C=O bond from carboxyl groups and C–C from ethers [39–41]. After functionalization (spectra 2 and 3), the number of absorption bands significantly increase as well as their intensity, which suggests that both phenolic compounds destroy the compact structure of PET waste and generate a much larger number of functional groups. Thus, the absorption bands from 1850–860 cm$^{-1}$ (aromatic ethers), 1720 cm$^{-1}$ (C=O from carboxyl groups), 1630 cm$^{-1}$ (C–O from carbonyl or carboxyl groups), 1544 cm$^{-1}$ (C–O of aromatic compounds ), 1191–1259 cm$^{-1}$ (asymmetric stretching of C–O–C) from spectra 2 and 3 clearly indicates that, after functionalization, such functional groups appear in the structure of obtained adsorbent materials and may represent the binding sites for Cu(II) ions from aqueous solution. This observation is sustained by the experimental data presented in previous sections, which show that after functionalization, the adsorption capacities of obtained adsorbent materials are significantly larger compared with PET waste (see Section 3.1). However, most of the absorption bands in spectra 2 and 3 from Figure 6 are split in 2, 3, or even 4 bands with close maximum absorption wave length (15–30 cm$^{-1}$). This suggests that most of the newly formed functional groups have different chemical vicinities and therefore not all are available for metal ion interactions. In this way, it can explain the relatively low spontaneity of Cu(II) ions adsorption processes on 8-Ph-PET and 2-OH-Ph-PET adsorbents. Therefore, this issue will need to be considered in subsequent studies to improve the adsorptive performances of materials obtained through the functionalization of PET waste.

## 4. Conclusions

In this study, the functionalization of PET waste was performed with three phenolic compounds (phenol, p-chloro-phenol, and hydroxyquinone), and the obtained materials were tested for the removal of Cu(II) ions from aqueous solution. The analysis of the obtained experimental results has shown that the functionalization of PET waste with phenol and hydroxyquinone leads to obtaining adsorbent materials with higher adsorption capacities than PET waste, while in the case of using p-chloro-phenol

for the functionalization, the obtained material has a lower adsorption capacity compared with PET waste. In addition, the values of the adsorption capacity also depend on the amount of phenolic compound used for functionalization. Thus, the adsorbent material obtained by the functionalization of PET waste with 8 g of phenol (8-Ph-PET) has the best adsorptive performances at low initial concentrations of Cu(II) ions, while the adsorbent material obtained by the functionalization of PET waste with 2 g of hydroxyquinone (2-OH-Ph-PET) is more efficient in removal of high concentrations of Cu(II) ions. The equilibrium data are well described by the Langmuir isotherm model, and the maximum adsorption capacities calculated for these two materials are 12.80 mg Cu(II) g$^{-1}$ for 8-Ph-PET and 61.73 mg Cu(II) g$^{-1}$ for 2-OH-Ph-PET, respectively. This significant increase in the adsorption capacities of 8-Ph-PET and 2-OH-Ph-PET adsorbents compared to PET waste is mainly due to the increase in the number of functional groups, as shown by FTIR spectra. However, the structural characteristics of these materials, together with kinetic and desorption studies, will be presented in detail in subsequent studies, to highlight the applicative potential of these materials in the processes of metal ion removal.

**Author Contributions:** Conceptualization, L.B.; Data curation, O.I.U. and D.B.; Formal analysis, O.I.U.; Investigation, L.B., A.M.M. and D.B.; Methodology, L.B., A.M.M. and D.B.; Project administration, L.B. All authors have read and agreed to the published version of the manuscript.

**Funding:** This research received no external funding.

**Conflicts of Interest:** The authors have declared no conflict of interest.

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
