# Peer review of "Functionalized PET Waste Based Low-Cost Adsorbents for Adsorptive Removal of Cu(II) Ions from Aqueous Media"

_water, doi:10.3390/w12092624_

Round 1

Reviewer 1 Report

Review:

Manuscript Number: water-918467

The authors reported “Low-cost adsorbent materials based on functionalized PET waste and their use in the removal of Cu(II) ions from aqueous media”. There are a number of things, however, that require improvement before it can be accepted. Following are my comments for improvements before this review article could be considered for publication.

  1. Check and revise the paper carefully, especially for the format errors.
  2. Try to improve the title. The title could be “Functionalized PET waste based low-cost adsorbents for adsorptive removal of Cu(II) ions from aqueous media”
  3. Improve the abstract. Spell out the “PET” at the first approach in the abstract.
  4. Include the purity of the purchased chemicals.
  5. Did the authors try to characterize the materials such as by SEM, EDS etc.
  6. The introduction needs to be improved. Cu ion is a water pollutant. Make a general paragraph on water pollutants and water treatment processes. There are so many pollutants like dyes, antibiotics, heavy metals etc. and so many wastewater treatment processes such as catalytic reduction, photocatalysis, Fenton process etc. Discuss them in the general paragraph and then verify why you choose Cu ion as a pollutant and adsorption as treatment method. The following recently published articles are suggested to cite. Applied Surface Science, 497 (2019) 43608; Catal. Comm, 130 (2019) 105753; J. Mol. Liq. 290 (2019) 111059.
  7. Section 2.3: The equation needs to have a proper reference. Science of the Total Environment, 698 (2020) 134214.
  8. Authors should carry out the kinetic studies for Cu(II) removal.
  9. Did the authors try temperature dependent study ?
  10. The conclusion needs to improve significantly. Also include some future remarks.

Author Response

Dear Reviewer,

            Thank you very much for your observations and recommendations. We read them very carefully and all have been considered in this variant of manuscript.

  1. The manuscript has been carefully checked, and I hope that now all spelling and grammar mistakes have been corrected.
  2. The title has been changed according with your recommendation.
  3. The abbreviation has been explained, and the abstract has been improved.
  4. In Experimental section was mentioned that “All chemical reagents were of analytical grade and were used without purification”.
  5. The obtained materials have been characterized using FTIR, SEM, EDX, TGA, but as a very detailed discussion is needed to interpret the obtained results, they will be included in another study.
  6. The introduction has been improved, according to your suggestions. Only one paper has been included in the references list, because the others do not have the same topic with this study.
  7. Section 2.3: Equation (1) is well known and it is not necessary to mention a reference.
  8. Kinetic studies have been performed, and their results will be included in a future study, along with a more detailed analysis of the structural characteristics of these materials.
  9. We are now working on studies on the influence of temperature.
  10. Conclusions have been revised.

.

            I hope that all these details are sufficiently to clarify the aspects underlined in the reviewer comments.

Best regards,

Prof.dr.habil.chem. Laura Bulgariu

Department of Environmental Engineering and Management

“Cristofor Simionescu” Faculty of Chemical Engineering and Environmental Protection

Technical University “Gheorghe Asachi” of IaÅŸi

Mangeron 71A, 700050-IaÅŸi, Romania

E-mail: lbulg@ch.tuiasi.ro / laurabul73@yaho.com

Reviewer 2 Report

Include errors associated to parameters in table 1.

Author Response

Dear Reviewer,

            Thank you very much for your observation.

  1. In the manuscript was mentioned that “All adsorption experiments were carried out in duplicate and the standard deviation, obtained from ANOVA statistical analysis, were lower than 1.5 %”.

I hope that all these details are sufficiently to clarify the aspects underlined in the reviewer comment.

Best regards,

Prof.dr.habil.chem. Laura Bulgariu

Department of Environmental Engineering and Management

“Cristofor Simionescu” Faculty of Chemical Engineering and Environmental Protection

Technical University “Gheorghe Asachi” of IaÅŸi

  1. Mangeron 71A, 700050-IaÅŸi, Romania

E-mail: lbulg@ch.tuiasi.ro / laurabul73@yaho.com

Reviewer 3 Report

see the attachment

Author Response

Dear Reviewer,

            Thank you very much for your observations and recommendations. We read them very carefully and all have been considered in this variant of manuscript.

  1. The manuscript has been carefully checked, and I hope that now all spelling and grammar mistakes have been corrected.
  2. Suggested papers have been added to the reference list, except for the first, which is not available online. We would be grateful if you could send us.
  3. The description of the methods has been improved. The average surface area of the PET flakes used in these experiments was added to the manuscript.
  4. Results and discussion: This section has been revised.
  5. Conclusions: This section has been improved and some future trends have been added.

I hope that all these details are sufficiently to clarify the aspects underlined in the reviewer comment.

Best regards,

Prof.dr.habil.chem. Laura Bulgariu

Department of Environmental Engineering and Management

“Cristofor Simionescu” Faculty of Chemical Engineering and Environmental Protection

Technical University “Gheorghe Asachi” of IaÅŸi

D. Mangeron 71A, 700050-IaÅŸi, Romania

E-mail: lbulg@ch.tuiasi.ro / laurabul73@yaho.com

Reviewer 4 Report

This manuscript reports the adsorption of copper ions from water using recycled PET waste. The removal of metal ions from water has been extensively covered in the literature and the present research does not provide neither new findings nor significant improvements over previously reported sorbents. The kinetic study is missing and the sorption mechanism is not fully discussed, in particular, the effects of important environmental factors (i.e. pH, temperature, ionic strength...) are lacking. While the adsorbent material is synthesized from plastic waste, its sorption performance is not competitive. For the above reasons (and the additional comments below), I recommend against the publication of this research unless extensive revisions/additions are undertaken by the authors.  

Additional comments:

  • Linearization isotherms models have been demonstrated inappropriate in predicting the goodness of fit [Foo et al Chem Eng J 2010, 156, 1, 2-10]. Therefore, non-linearized forms of the Langmuir and Freundlich isotherm models should be utilized. 
  • Although it is mentioned in Table 2, the solution pH is not reported in the experimental section. Dichiara et al [ACS Appl Mater Interfaces, 2015, 7, 28, 15674] demonstrated the critical importance of the solution pH on the adsorption of cations, especially when oxygen moieties are present on the sorbent surface (i.e. hydroxyl and carboxyl groups). Given that Coulombic interactions likely govern the sorption mechanism for the functionalized PET, the effect of the solution pH must be investigated. 
  • Study the influence of solution pH may also provide useful information about the desorption process to possibly recover valuable copper ions. In this regard, the regeneration efficiency over several sorption cycles may be examined to make the PET sorbent more appealing for practical applications.     
  • Error bars should be reported from Figure 1 through 4. 
  • Additional characterization of the sorbent material is needed: electron microscopy, specific surface area, pore size distribution... Currently, the material has only been characterized by FTIR spectroscopy. 
  • The authors should describe the rationale for selecting the references in Table 2 to compare their sorption data. It would make sense to compare the data with other sorbents prepared from recycled resources, otherwise there exist materials (i.e. graphene...) with much greater uptake capacities than those listed in Table 2 [Dichiara et al [ACS Appl Mater Interfaces, 2015, 7, 28, 15674].

Author Response

Dear Reviewer,

            Thank you very much for your observations and recommendations. We read them very carefully and all have been considered in this variant of manuscript.

  1. It is true that the removal of metal ions by adsorption is a widely studied issue, but the novelty of this study is the use as adsorbents of materials obtained from PET waste. This is a major advantage, given the current problems related to the disposal of PET waste. This aspect is mentioned in the manuscript.
  2. Kinetic and optimization studies have been performed and their results will be included in a future study, together with a more detailed analysis of the structural characteristics of these materials. This is because these results require a more detailed discussion for their interpretation. In this manuscript is highlighted only the potential applicability of these functionalized adsorbents.
  3. “While the adsorbent material is synthesized from plastic waste, its sorption performance is not competitive.” Why??? I am not agree. Such functionalizations that can turn plastic waste into a competitive adsorbent.
  4. It is true. I know this paper. But in this manuscript we use the liniarized equation of Langmuir model to calculate the maximum adsorption capacities and to highlight the good correspondence between experimental and calculated values. Because of this, we prefer that Langmuir representations remain in linearized form.
  5. Initial solution pH has been mentioned in Experimental section. The influence of this parameter on the Cu(II) adsorption efficiency will be detailed discussed in a future study, as well as desorption results.
  6. It is mentioned now in the manuscript that: “All adsorption experiments were carried out in duplicate and the standard deviation, obtained from ANOVA statistical analysis, were lower than 1.5 %”, to avoid the addition of error bars in figure.
  7. The obtained materials have been characterized using FTIR, SEM, EDX, TGA, but as a very detailed discussion is needed to interpret the obtained results, they will be included in another study.
  8. The adsorbent presented in table 2 were selected considering: (i) their simple preparation, and (ii) their availability. That is why we did not mention here the functionalized materials and nanomaterials, even if their efficiency in the processes of Cu (II) ion removal is clearly superior.

I hope that all these details are sufficiently to clarify the aspects underlined in the reviewer comment.

Best regards,

Prof.dr.habil.chem. Laura Bulgariu

Department of Environmental Engineering and Management

“Cristofor Simionescu” Faculty of Chemical Engineering and Environmental Protection

Technical University “Gheorghe Asachi” of IaÅŸi

D. Mangeron 71A, 700050-IaÅŸi, Romania

E-mail: lbulg@ch.tuiasi.ro / laurabul73@yaho.com

Reviewer 5 Report

No comments 

Author Response

Dear Reviewer,

            Thank you very much for your appreciation. The manuscript has been carefully checked, and I hope that now all spelling and grammar mistakes have been corrected.

Best regards,

Prof.dr.habil.chem. Laura Bulgariu

Department of Environmental Engineering and Management

“Cristofor Simionescu” Faculty of Chemical Engineering and Environmental Protection

Technical University “Gheorghe Asachi” of IaÅŸi

D. Mangeron 71A, 700050-IaÅŸi, Romania

E-mail: lbulg@ch.tuiasi.ro / laurabul73@yaho.com

Round 2

Reviewer 1 Report

The manuscript has been improved as suggested.

Author Response

Dear Reviewer,

            Thank you for your time and effort in reviewing our manuscript.

Best regards,

Prof.dr.habil.chem. Laura Bulgariu

Department of Environmental Engineering and Management

“Cristofor Simionescu” Faculty of Chemical Engineering and Environmental Protection

Technical University “Gheorghe Asachi” of IaÅŸi

D. Mangeron 71A, 700050-IaÅŸi, Romania

E-mail: lbulg@ch.tuiasi.ro / laurabul73@yaho.com

Reviewer 4 Report

While the authors addressed the reviewers comments, it seems that most of the revisions will be made available in an upcoming study following up the research described in the present manuscript. Therefore, there is not much added value in this revised manuscript (i.e. characterization, kinetics, desorption and recycling). Nevertheless, this study is worthy of interest for the readership of Water and I recommend for its publication after the authors add further research from the literature about adsorbents synthesized from waste in their introduction and in Table 2 (i.e. sorbents from waste cigarette filters, fruit peel, sea shell). 

Author Response

Dear Reviewer,

            Thank you very much for your recommendations. Table 2 was completed with several adsorbent materials synthesized from waste, but we considered only the studies in which the retention of Cu (II) ions was performed under similar experimental conditions.

Best regards,

Prof.dr.habil.chem. Laura Bulgariu

Department of Environmental Engineering and Management

“Cristofor Simionescu” Faculty of Chemical Engineering and Environmental Protection

Technical University “Gheorghe Asachi” of IaÅŸi

D. Mangeron 71A, 700050-IaÅŸi, Romania

E-mail: lbulg@ch.tuiasi.ro / laurabul73@yaho.com